# A Dedicated Mycosis Flask Increases the Likelihood of Identifying Candidemia Sepsis

**DOI:** 10.3390/jof9040441

**Published:** 2023-04-04

**Authors:** Magnus G. Ahlström, Valeria S. Antsupova, Michael Pedersen, Helle Krogh Johansen, Dennis Schrøder Hansen, Inge Jenny Dahl Knudsen

**Affiliations:** 1Department of Clinical Microbiology, Herlev & Gentofte Hospital, 2730 Herlev, Denmark; 2Department of Clinical Microbiology, Amager & Hvidovre Hospital, 2650 Hvidovre, Denmark; 3Department of Clinical Microbiology, Rigshospitalet, 2100 Copenhagen, Denmark; 4Department of Clinical Medicine, Faculty of Health and Medical Sciences, University of Copenhagen, 1165 Copenhagen, Denmark

**Keywords:** candidemia, blood culture, time to detection, mycosis flask, sepsis

## Abstract

Introduction: Candidemia is a severe condition associated with high mortality, and fungi are often not covered by empiric antimicrobial regimes for sepsis. Therefore, the shortest possible time to detection of yeast in the blood is of the essence. Materials and methods: We performed a cohort study of blood culture flasks drawn from patients aged 18 or older in the capital region of Denmark. In 2018 a blood cultures set consisted of two aerobic and two anaerobic flasks. This was changed in 2020 to two aerobic, one anaerobic, and one mycosis flask. We used time-to-event statistics to model time to positivity and compared 2018 with 2020; further, we stratified analyses on the blood culture system used (BacTAlert™ vs. BACTEC™) and high-risk vs. low-risk departments. Results: We included 175,416 blood culture sets and 107,077 unique patients. We found an absolute difference in the likelihood of identifying fungi in a blood culture set of 1.2 (95% CI: 0.72; 1.6) pr. 1.000 blood culture sets corresponding to the number needed to treat 853 (617; 1382). In high-risk departments, the absolute difference was profound, whereas it was negligible and statistically non-significant in low-risk departments 5.2 (95% CI: 3.4; 7.1) vs. 0.16 (−0.17; 0.48) pr. 1.000 blood culture sets. Conclusions: We found that including a mycosis flask in a blood culture set increases the likelihood of identifying candidemia. The effect was mainly seen in high-risk departments.

## 1. Introduction

Invasive fungal infections (IFI) such as candidemia are often seen in otherwise severely ill patients and have a high mortality rate depending on the underlying condition of the patient [1,2]. Even in the group of patients were candidemia, is most often detected, it is still a relatively rare etiology of blood stream infection. Antifungals are therefore usually not included in empiric antimicrobial treatment. As with bacteraemia, the time to initiation of sufficient antimicrobial treatment is critical for the survival of the patient [1,2,3]. An optimized diagnosis of candidemia is warranted.

Blood culturing is still a cornerstone in the diagnosis of candidemia and other IFI, but the sensitivity for detecting candidemia can also be affected by concomitant bacterial growth [4]. The special media for improving growth in blood culture flasks and thereby decreasing time to detection of candidemia has been tested in simulated settings, with or without antifungals [5,6], and also in cross-sectional studies [7]. In the current study, we present data from a large comparative cohort study.

The main purpose of the current study was to evaluate whether the inclusion of the mycosis flask in a blood culture set increased the likelihood of detecting patients with candidemia. Secondary purposes were to (1) evaluate whether the effect of including the mycosis flask differed for high-risk departments compared to low-risk departments and (2) to evaluate whether the change in blood culturing system from BacTAlert™ (bioMerieux, Marcy-l’Étoile, France) to BACTEC™ (BD, Mississauga, ON, Canada) affected the likelihood of identifying candidemia.

## 2. Materials and Methods

### 2.1. Study Population

We performed a cohort study of all blood culture flasks drawn from patients aged 18 or older in the capital region of Denmark (approximately 1.8 mill inhabitants). Blood culture sets were incubated and analyzed at the three Departments of Clinical Microbiology (DCM) in the capital region of Denmark, i.e., Herlev DCM, Hvidovre DCM, and Rigshospitalet DCM. We compared the years 2018 and 2020 (1 February 2020, to 31 January 2021, for Rigshospitalet because of the switch to a new blood culture system in 2021 to ensure one year of follow-up-hereafter called 2020). Because of a change in policy, the components of a blood culture set changed, i.e., Before 2020 a blood culture set consisted of 2 aerobic and 2 anaerobic blood culture flasks. This changed to two aerobic, one anaerobic, and one mycosis flask in 2020. Because of this, all DCMs that used the BacTAlert™ system in 2018 (Appendix A) switched to BACTEC™ in 2020 to comply with the new standards. These departments accounted for approximately half of the blood culture flasks analyzed.

### 2.2. Data Extraction

Data were extracted from each DCMs electronic laboratory information system. All Danish citizens are given a unique identification number at birth or immigration, and all tourists are given a surrogate unique identification number when admitted to a hospital. More than 99% of all hospital admissions and blood culturing procedures are in tax-financed hospitals, free of direct patient charge.

We excluded flasks and individuals with no unique identification number, quality controls, and environmental samples. Furthermore, we excluded flasks without a flask type registered.

### 2.3. Blood Culture Processing

After collection, the blood culture flasks were transported to the DCM and incubated as fast as possible. At hospitals where there was no DCM, there are satellite blood culture cabinets at which the flasks can be incubated. The cultures are incubated at 35 °C. The growth of microorganisms in the flask is detected by the production of CO_2_. When a flask is positive, trained laboratory technicians perform Gram staining and light microscopy and species identification via Matrix-assisted laser desorption-ionization time of flight (MALDI-TOF) (Bruker, Bremen, Germany) directly on the blood in the positive flask.

### 2.4. Definitions

The primary outcome was the detection of fungi in a blood culture flask or blood culture set when analyzing blood culture sets. Time to detection (TTD) was calculated as the time from incubation of the flask/set until the system detected growth. For each flask, the growth of any bacteria/fungus was noted. If a flask had growth of both bacteria and any fungi, this was also noted. If the blood culture was negative in microscopy but later showed growth of fungi on the agar plates, the time of the first registration of an isolate was used instead of time until the system detected growth. Flask type was divided into aerobic, anaerobic, and mycosis.

A blood culture set was defined as all blood culture flasks drawn from a patient no more than 3 h apart and consisting of at least 3 blood culture flasks. For blood culture sets that were only positive for fungi or positive for both fungi and bacteria, TTD was defined as the earliest TTD for a flask positive for fungi. For blood culture sets that were only positive for bacteria, TTD was defined as the earliest TTD. We defined high-risk departments as any of the following departments (Department of Hematology, Department of Oncology, Gastric surgery, Gastroenterology, Department of Urology, Department of Nephrology, and Intensive Care Unit) as these departments had the highest incidence of candidemia in our cohort, low-risk departments were defined as all other departments. Blood culture flasks and blood culture sets were also divided according to whether they were part of the BacTAlert™ or the BACTEC™ system as described above.

### 2.5. Statistical Methods

We used cumulative incidence function curves in a multistate competing risks analysis to visualize the TTD of fungi. The model had 3 states (culture negative, positive for bacteria, and positive for fungi). If the culture set was positive for both bacteria and fungi, it was categorized as positive for fungi and was not allowed to change state. The growth of a bacterium was considered a competing risk. First, we analyzed individual blood culture flasks to evaluate whether the mycosis flask was better at identifying candidemia. In order to evaluate the direct clinical impact, we also analyzed blood culture sets and compared blood culture sets analyzed in 2018 with culture sets analyzed in 2020 to evaluate whether the inclusion of a mycosis flask had a substantial clinical impact. We did sensitivity analyses where we compared blood culture sets with a mycosis flask included with blood culture sets with no mycosis flask included. The model violated the proportional hazards assumption (visually evaluated from complementary log–log curves), which is why we did not do any regression analyses on the data. We used the log-rank test to test for differences between curves. From the models, we calculated the absolute risk difference (ARD) and calculated the numbers needed to treat (NNT) via the formula NNT = 1/ARD. Among the high-risk departments, we did stratified analyses on all department types.

The extraction of data from electronic laboratory information systems was approved by the local data protection agency (journal no. P-2021-598) and the Centre of Regional Development (journal no. R-21057173).

Statistical analyses were performed using R version 3·6·0 (R Development Core Team 2018). R: A language and environment for statistical computing. R Foundation for Statistical Computing, Vienna, Austria).

## 3. Results

A total of 737,132 blood culture flasks were included in the study, 347,722 and 389,410 blood culture flasks in 2018 and 2020, respectively. A total of 107,077 patients were included: 52,215 in 2018 and 54,862 in 2020. In 2018 almost no mycosis flasks were used, and a blood culture set consisted primarily of 2 aerobic and 2 anaerobic flasks, as was recommended by the local DCM. As expected, this changed in 2020 to approximately 25% mycosis flasks, 50% aerobic flasks, and 25% anaerobic flasks, as was the advocated policy in our region. This was supported by the electronic patient medical record in the capital region of Denmark, which is an EPIC™-based system (EPIC, Wisconsin, USA). The fungi identified were mainly different *Candida* species, primarily *C. albicans* (128 (36%) in 2018 and 238 (41%) in 2020) and *C. glabrata* (145 (41%) in 2018 and 228 (40%) in 2020). The hospitals that used the BACTEC™ system in both 2018 and 2020 had a small (13%) increase in the positive rate of blood culture flasks for fungi from 0.15% to 0.17%. However, the hospitals that changed from the BacTAlert™ to the BACTEC™ system had a substantial increase (54%) in the positive rate, 0.06% in 2018 compared with 0.13% in 2020 (Table 1).

For the analyses of blood culture sets, we included 175,416 full blood culture sets from 104,495 patients (83,159 in 2018 and 92,257 in 2020) (Table 2). A total of 50,251 and 54,244 unique patients had at least one full blood culture set performed in 2018 and 2020, respectively. Most blood culture sets consisted of 4 flasks, i.e., 74,643 (90%) and 84,027 (91%) in 2018 and 2020, respectively.

Figure 1 shows the cumulative incidence function curves of the TTD of fungi in individual blood culture flasks stratified on flask type. We found that for the mycosis flasks, 0.22% were positive for fungi. For aerobic flasks, the number was 0.16%, and for anaerobic flasks, 0.04%. The graph also shows that they generally detect fungi faster than the corresponding aerobic and anaerobic flasks (Figure 1). This implies that the mycosis flask is substantially better at detecting fungi than both aerobic flasks and anaerobic flasks. This was mirrored when we analyzed the TTD of full blood culture sets, which is a more clinically relevant measure than individual blood culture flasks. The overall estimate showed a higher proportion of blood culture sets being positive in 2020 compared with 2018 (see Figure 2a and Table 1 and Table 3). In low-risk departments, we found a negligible and non-statistically significant difference between 2018 and 2020, risk difference (0.16 (95% CI: −0.17–0.48, NNT = 6367), pr. 1000 blood culture flask). When we analyzed high-risk departments, we found a highly statistically significant absolute risk difference of 5.2 (95% CI: 3.4–7.1, NTT = 191), pr. 1000 blood culture flask.

In stratified analyses on high-risk departments, we found that in the Department of Oncology and the Department of Gastroenterology, the effect had diminished and was, in fact, negative; however, the numbers are small. In the Department of Haematology, Department of Gastric surgery, and the Intensive Care Unit, the effect was large and statistically significant even in the stratified analyses.

In the analyses stratified on the hospitals that used BacTAlert™ vs. BACTEC™ in 2018, we found that the BACTEC™ blood culture system had a substantially higher positive rate than the BacTAlert™ system, i.e., an absolute difference of 1.6 (95% CI: 1.0; 2.1) pr. 1000 blood culture set corresponding to an NNT of 637 (95% CI: 478; 953), which implies that the BACTEC™ system is better than the BacTAlert™ system at identifying candidemia (Table 3). The most interesting comparison was the analyses of the hospitals that used the BACTEC™ system in both 2018 and 2020 since differences here can be attributed to the change in the composition of the blood culture set. In the overall analysis, we found a non-statistically significant change of 0.62 (95% CI: −0.17; 1.4); however, in high-risk departments, there was a large and statistically significant difference of 2.7 (95% CI: 0.56; 4.9, NNT = 364) pr. 1,000 blood culture set; however, in low-risk departments, the difference was not statistically significant and actually trended toward being negative −0.42 (95% CI: −1.0; 0.19). For the hospital that changed from BacTAlert™ to BACTEC, the difference was even more pronounced. In high-risk departments, we found a difference of 9.7 (95% CI: 6.4–13, NNT = 103) pr. 1000 blood culture set and in low-risk departments 0.48 (95% CI: 0.1–0.86, NNT = 2075) pr. 1000 blood culture set. All in all, these analyses imply that the change from BacTAlert™ to BACTEC™ had a pronounced effect on the number of candidemia detected, as does the inclusion of the mycosis flask in a blood culture set.

This section may be divided into subheadings. It should provide a concise and precise description of the experimental results, their interpretation, as well as the experimental conclusions that can be drawn.

## 4. Discussion

In this large cohort study based on real-life data of cultures, we found that the inclusion of a mycosis flask in all blood culture sets increased the likelihood of identifying candidemia in patients. The overall effect was moderate; however, in patients from high-risk departments and in departments that switched from the BacTAlert™ system to the BACTEC™ system, the effect was profound.

It seems intuitive that including a blood culture flask dedicated to detecting fungi would increase the likelihood of identifying fungi in a blood culture set. Several studies have compared the efficacy of the mycosis flask with conventional aerobic flasks and found the mycosis flask to be superior [6,8,9]. Zheng et al. found that including a mycosis bottle in the blood culture set yielded an increased detection rate of 22.6–24.3%, depending on the type of analysis [7]. This is consistent with the results of our study. Surprisingly we found that the effect in low-risk departments was substantially lower than in high-risk departments, both with regard to the absolute difference and the relative difference, implying statistical interaction. Whether or not this is a chance finding or, indeed, a phenomenon that can be reproduced needs to be evaluated in future studies. The implication of such findings is that the inclusion of the mycosis flask in blood culture sets from low-risk departments is not indicated.

In our study, we found that the BACTEC™ system detected fungi better than the BacTAlert™ system. One comparative study found that the BACTEC™ system had statistically significantly lower TTD than the BacTAlert™ system for the detection of fungi, which supports the finding in the current study [10]. Data in the current study suggests that this is not only true for fungi but also for bacteria. Because this was not in the scope of this work, the subject has not been evaluated in depth; however, other studies have found similar results [11,12], but conflicting data have also been published [10].

Studies that have evaluated whether it is cost effective to include a mycosis flask in a blood culture set are few, and the studies have not evaluated the effect in a whole population but rather in selected cases with known candidemia [8,9,13], and in some cases, the fungi have been added as a culture, i.e., not from a clinical sample [5,6]. With the data from this study, we present a benchmark on what to expect with regard to the number of positive blood culture sets. In addition, we provide estimates of NNT. For the departments that used the BACTEC™ system both before and after the change in the composition of the blood culture set, we found an NNT of 364 in high-risk departments, i.e., 364 patients need to be blood cultured with the new blood culture set to find one extra case of candidemia as compared with the old blood culture set. For the hospitals that also changed the blood culture system, this number was 103. This is a low number, i.e., a substantial effect, given that candidemia is a relatively rare condition even in high-risk departments.

The policy change that led to the change in the composition of the blood culture set in our region meant that one anaerobic blood culture flask was replaced by a mycosis flask. In the current setting, two non-lytic anaerobic bottles were replaced by one BACTEC-Lytic bottle. Indeed this blood culture flask has been shown to be superior/non-inferior to other anaerobic blood culture flasks [14]; however, some concern still exists about whether one lytic flask is sufficient at detecting bacteremia with strict anaerobic bacteria. Another concern when we implemented this change was that previously approximately 32–40 mL (8–10 mL pr flask) of blood was drawn from each individual for detection of bacteria (4 flasks of 8–10 mL), but since bacteria are usually not detected in the mycosis flask only 24–30 mL was drawn for each patient after the change. This is still within the range of the internationally recommended 20–30 mL [15]; nonetheless, this could affect the sensitivity of blood culturing for the detection of bacteria. We have not evaluated how the change affected the growth of anaerobic bacteria in the current study; however, overall bacterial growth does not seem to be affected by the change in the composition of the blood culture set. Analyses regarding the loss in sensitivity of strictly anaerobic and facultative anaerobic bacteria are warranted and need to be conducted in a separate study. The current study group is planning to conduct these analyses in the future.

The study has some major strengths. The study included more than 700.000 blood culture flasks, which made it possible to study a relatively rare event with sufficient power. The intervention was introduced as the standard of practice, which made the risk of confounding by indication very small. The population-based design and the use of the unique Danish identification number made it possible to track individuals across databases and hospitals with very high accuracy.

The study has some limitations. We wanted to study one intervention, i.e., replacing an anaerobic flask in a blood culture set with a mycosis flask; however, the switch from the BacTAlert™ system to the BACTEC™ system also showed to have a substantial effect on the number of positive blood culture sets, however since the composition of the patient populations in the two groups is almost constant over the two periods, we believe that we are able to observe the pure effect of the intervention in the hospitals that only used the BACTEC™ system. The study by design was not a clinical randomized trial. We, therefore, cannot exclude confounding factors to contribute to the effects observed in this study. This could be the drift of patients from high-risk departments out of the region or vice versa. We did not have information on the amount of blood in each culture flask; however, we have no reason to believe that this was different in the two periods. Finally, we did not have any clinical data available and therefore could not evaluate whether the intervention was associated with a more favorable outcome, e.g., with regard to the EQUAL score [16].

## 5. Conclusions

Including a mycosis flask in a blood culture set substantially increases the likelihood of identifying candidemia. The effect is mainly seen in high-risk departments.

## Figures and Tables

**Figure 1 jof-09-00441-f001:**
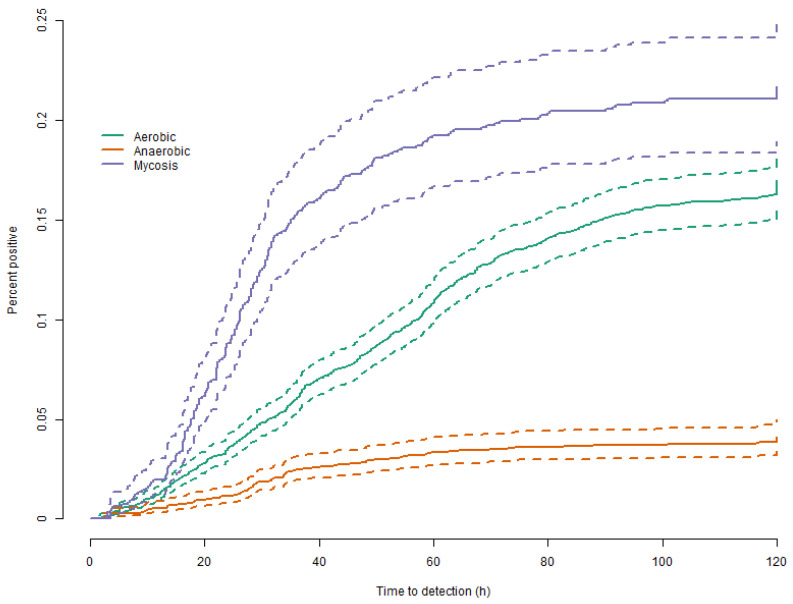
Time to detection of fungi-blood culture flasks: The figure shows the cumulative incidence function curves for the time to detection of fungi in each individual flask included in the study as a function of time. We found that the mycosis flasks had a lower time to detection, i.e., detected fungi faster than both aerobic and anaerobic flasks, and that the proportion of positive flasks was higher app. 0.22% for the mycosis flasks compared to appr. 0.17% for the aerobic flasks and 0.04% for the anaerobic flasks.

**Figure 2 jof-09-00441-f002:**
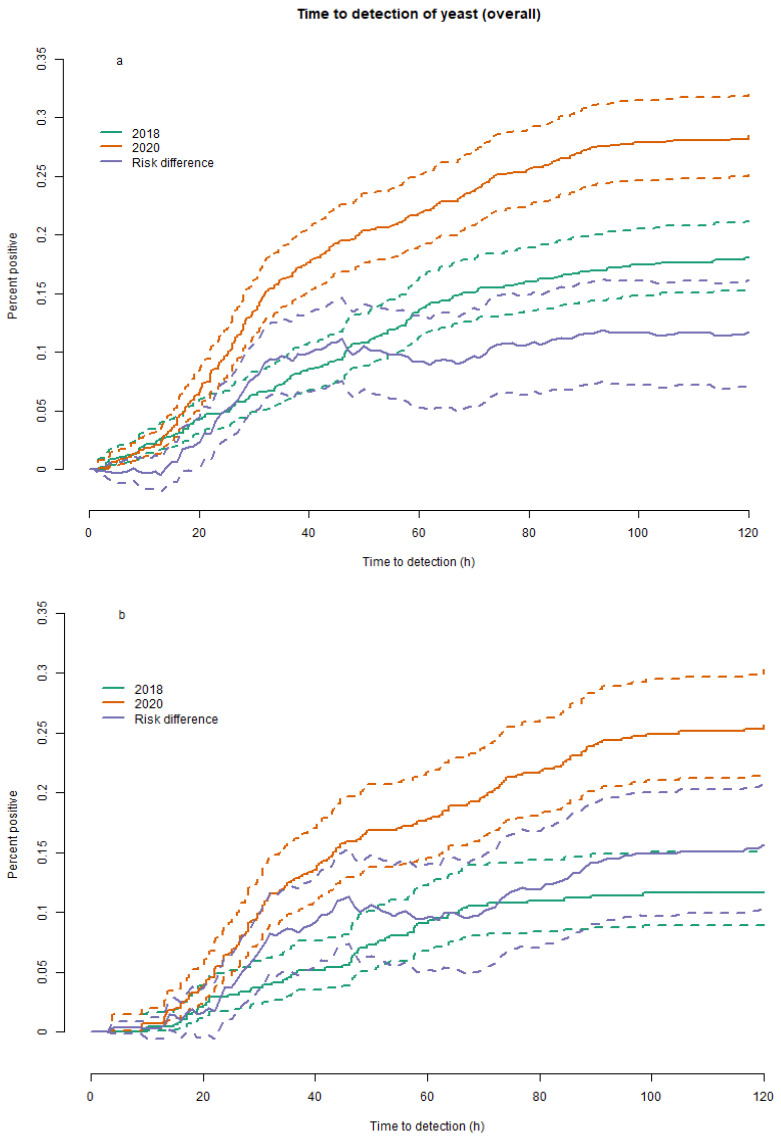
Time to detection of fungi-blood culture sets. The figure shows the cumulative incidence function curves with 95% confidence intervals for blood culture sets drawn in 2018 (before the implementation of dedicated mycosis flasks) and in 2020 (after the implementation of dedicated mycosis flasks) and the absolute difference between the two estimates. Figure (**a**) is the overall estimates for all blood culture sets with yeast, and Figure (**b**) is the overall estimates for all blood culture sets from hospitals that used the BacTAlert™ system in 2018, in 2020, the BACTEC™ system was used at these hospitals, (**c**) shows the estimates for low-risk departments for all blood culture sets from hospitals that used the BacTAlert™ system in 2018 (**d**) shows the estimates for high-risk departments for all blood culture sets from hospitals that used the BacTAlert™ system in 2018, (**e**) shows the overall estimates for all blood culture sets from hospitals that used the BACTEC™ system in 2018, these hospitals also used the BACTEC™ system in 2020, (**f**) shows the estimates for low-risk departments for all blood culture sets from hospitals that used the BACTEC™ system in 2018, (**g**) shows the estimates for high-risk departments for all blood culture sets from hospitals that used the BACTEC™ system in 2018.

**Table 1 jof-09-00441-t001:** Characteristics of the study population individual flasks.

	2018	2018	2018	2020	2020	2020
	BACTEC™	BacTAlert™	Total	BACTEC™	BACTEC™ in the Hospital’s Formerly Using BacTAlert™ *^, #^	Total
Blood culture flasks *	155,450; 233 (0.15%)	192,272; 120 (0.06%)	347,722; 353 (0.1%)	162,245; 274 (0.17%)	227,165; 301 (0.13%)	389,410; 575 (0.15%)
Unique patients	22,346	30,808	52,215	22,383	33,371	54,862
Flask type *						
Mycosis flask	2409; 7 (0.29%)	0; 0 (-)	2409; 7 (0.29%)	38,659; 101 (0.26%)	56,765; 104 (0.18%)	95,424; 205 (0.21%)
Aerobic flask	76,511; 196 (0.26%)	96,027; 91 (0.09%)	172,538; 287 (0.17%)	80,450; 151 (0.19%)	113,457; 170 (0.15%)	193,907; 321 (0.17%)
Anaerobic flask	76,530; 30 (0.04%)	96,245; 29 (0.03%)	172,775; 59 (0.03%)	43,136; 22 (0.05%)	56,943; 27 (0.05%)	100,079; 49 (0.05%)
Department of Clinical Microbiology *						
Hvidovre	0; 0 (-)	125,515; 115 (0.09%)	125,515; 115 (0.09%)	0; 0 (-)	151,053; 212 (0.14%)	151,053; 212 (0.14%)
Herlev	108,671; 76 (0.07%)	66,757; 5 (0.01%)	175,428; 81 (0.05%)	113,688; 71 (0.06%)	76,112; 89 (0.12%)	189,800; 160 (0.08%)
Rigshospitalet	46,779; 157 (0.34%)	0; 0 (-)	46,779; 157 (0.34%)	48,557; 203 (0.42%)	0; 0 (-)	48,557; 203 (0.42%)
Department *						
Low risk	102,452; 61 (0.06%)	167,998; 58 (0.03%)	270,450; 119 (0.04%)	107,084; 43 (0.04%)	194,647; 109 (0.06%)	301,731; 152 (0.05%)
High risk	52,998; 172 (0.32%)	24,274; 62 (0.26%)	77,272; 234 (0.3%)	55,161; 231 (0.42%)	32,518; 192 (0.59%)	87,679; 423 (0.48%)
Hematology	11,704; 6 (0.05%)	16; 0 (0%)	11,720; 6 (0.05%)	10,312; 21 (0.2%)	16; 0 (0%)	10,328; 21 (0.2%)
Oncology	9288; 11 (0.12%)	1105; 0 (0%)	10,393; 11 (0.11%)	9501; 5 (0.05%)	875; 0 (0%)	10,376; 5 (0.05%)
Gastric surgery	4319; 17 (0.39%)	10,086; 14 (0.14%)	14,405; 31 (0.22%)	5206; 49 (0.94%)	11,841; 45 (0.38%)	17,047; 94 (0.55%)
Gastroenterology	6626; 34 (0.51%)	3148; 8 (0.25%)	9774; 42 (0.43%)	5976; 32 (0.54%)	3492; 35 (1%)	9468; 67 (0.71%)
Urology	4180; 16 (0.38%)	0; 0 (-)	4180; 16 (0.38%)	3775; 25 (0.66%)	4; 0 (0%)	3779; 25 (0.66%)
Nephrology	8573; 9 (0.1%)	2055; 0 (0%)	10,628; 9 (0.08%)	7575; 15 (0.2%)	1908; 5 (0.26%)	9483; 20 (0.21%)
Intensive care unit *	8308; 79 (0.95%)	7864; 40 (0.51%)	16,172; 119 (0.74%)	12,816; 84 (0.66%)	14,382; 107 (0.74%)	27,198; 191 (0.7%)
Microscopy negative *	707; 9 (1.27%)	1243; 11 (0.88%)	1950; 20 (1.03%)	545; 7 (1.28%)	1234; 32 (2.59%)	1779; 39 (2.19%)
Candidemia–which species						
Candida albicans	82 (35%)	46 (38%)	128 (36%)	126 (46%)	112 (37%)	238 (41%)
Candida glabrata	82 (35%)	63 (52%)	145 (41%)	90 (33%)	138 (46%)	228 (40%)
Candida krusei	8 (3.4%)	0 (0%)	8 (2.3%)	4 (1.5%)	0 (0%)	4 (0.7%)
Candida parapsilosis	14 (6%)	4 (3.3%)	18 (5.1%)	28 (10%)	14 (4.7%)	42 (7.3%)
Candida tropicalis	11 (4.7%)	6 (5%)	17 (4.8%)	17 (6.2%)	13 (4.3%)	30 (5.2%)
Candida dubliniensis	18 (7.7%)	1 (0.83%)	19 (5.4%)	5 (1.8%)	10 (3.3%)	15 (2.6%)
Candida kefyr	13 (5.6%)	0 (0%)	13 (3.7%)	1 (0.36%)	7 (2.3%)	8 (1.4%)
Candida guilliermondii	0 (0%)	0 (0%)	0 (0%)	8 (2.9%)	6 (2%)	14 (2.4%)
Cryptococcus neoformans	0 (0%)	0 (0%)	0 (0%)	0 (0%)	2 (0.66%)	2 (0.35%)
Other fungi	11 (4.7%)	0 (0%)	11 (3.1%)	0 (0%)	1 (0.33%)	1 (0.17%)
Bacteraemia	6291 (4%)	7468 (3.9%)	13,759 (4%)	5508 (3.4%)	10,551 (4.6%)	16,059 (4.1%)
Combined bacteraemia and fungaemia	5 (0.003%)	15 (0.008%)	20 (0.006%)	1 (0.001%)	11 (0.005%)	12 (0.003%)

* All numbers are: Total number; Positive for any fungi (%positive), unless stated otherwise. # For detailed information on which centres used the BacTAlert™ system and the BACTEC™ system see Appendix A.

**Table 2 jof-09-00441-t002:** Characteristics blood culture sets (≥3 blood culture flasks taken no more than 3 h apart from the same patient).

	2018			2020		
	BACTEC™	BacTAlert™	Total	BACTEC™	BACTEC™ in the Hospital’s Formerly Using BacTAlert™ *	Total
Blood culture sets	34,977; 81 (0.23%)	48,182; 44 (0.09%)	83,159; 125 (0.15%)	37,724; 105 (0.28%)	54,533; 108 (0.2%)	92,257; 213 (0.23%)
Unique patients	20,996	30,155	50,251	21,825	33,304	54,244
Department of clinical microbiology						
Hvidovre	0; 0 (-)	31,883; 43 (0.13%)	31,883; 43 (0.13%)	0; 0 (-)	36,147; 71 (0.2%)	36,147; 71 (0.2%)
Herlev	26,142; 37 (0.14%)	16,299; 1 (0.01%)	42,441; 38 (0.09%)	27,590; 31 (0.11%)	18,386; 37 (0.2%)	45,976; 68 (0.15%)
Rigshospitalet	8835; 44 (0.5%)	0; 0 (-)	8835; 44 (0.5%)	10,134; 74 (0.73%)	0; 0 (-)	10,134; 74 (0.73%)
Number of flasks in blood culture set						
3	790; 2 (0.25%)	1470; 1 (0.07%)	2260; 3 (0.13%)	1361; 18 (1.32%)	611; 0 (0%)	1972; 18 (0.91%)
4	30,742; 62 (0.2%)	44,730; 35 (0.08%)	75,472; 97 (0.13%)	33,617; 60 (0.18%)	51,070; 70 (0.14%)	84,687; 130 (0.15%)
5	1624; 4 (0.25%)	34; 1 (2.94%)	1658; 5 (0.3%)	377; 3 (0.8%)	401; 3 (0.75%)	778; 6 (0.77%)
6	429; 6 (1.4%)	411; 0 (0%)	840; 6 (0.71%)	317; 2 (0.63%)	148; 2 (1.35%)	465; 4 (0.86%)
7	87; 0 (0%)	41; 0 (0%)	128; 0 (0%)	167; 1 (0.6%)	128; 4 (3.12%)	295; 5 (1.69%)
8	1052; 6 (0.57%)	1360; 7 (0.51%)	2412; 13 (0.54%)	1676; 20 (1.19%)	1873; 27 (1.44%)	3549; 47 (1.32%)
>8	1305; 7 (0.54%)	1496; 7 (0.47%)	2801; 14 (0.5%)	1885; 21 (1.11%)	2175; 29 (1.33%)	4060; 50 (1.23%)
Departments						
Low risk	24,547; 30 (0.12%)	42,212; 20 (0.05%)	66,759; 50 (0.07%)	25,964; 22 (0.08%)	47,804; 38 (0.08%)	73,768; 60 (0.08%)
High risk	10,430; 51 (0.49%)	5970; 24 (0.4%)	16,400; 75 (0.46%)	11,760; 83 (0.71%)	6729; 70 (1.04%)	18,489; 153 (0.83%)
Hematology	1697; 2 (0.12%)	4; 0 (0%)	1701; 2 (0.12%)	1783; 10 (0.56%)	4; 0 (0%)	1787; 10 (0.56%)
Oncology	2200; 5 (0.23%)	270; 0 (0%)	2470; 5 (0.2%)	2235; 1 (0.04%)	209; 0 (0%)	2444; 1 (0.04%)
Gastric surgery	925; 8 (0.86%)	2692; 3 (0.11%)	3617; 11 (0.3%)	1166; 13 (1.11%)	2878; 16 (0.56%)	4044; 29 (0.72%)
Gastroenterology	1420; 14 (0.99%)	820; 4 (0.49%)	2240; 18 (0.8%)	1305; 7 (0.54%)	821; 9 (1.1%)	2126; 16 (0.75%)
Urology	1000; 7 (0.7%)	0; 0 (-)	1000; 7 (0.7%)	920; 11 (1.2%)	1; 0 (0%)	921; 11 (1.19%)
Nephrology	1967; 4 (0.4%)	506; 0 (-)	2473; 4 (0.4%)	1717; 7 (0.76%)	464; 3 (300%)	2181; 10 (1.09%)
Intensive care unit	1221; 11 (0.9%)	1678; 17 (1.01%)	2899; 28 (0.97%)	2634; 34 (1.29%)	2352; 42 (1.79%)	4986; 76 (1.52%)
Bacteremia	3396 (9.7%)	3101 (6.4%)	6497 (7.8%)	3797 (10%)	5517 (10%)	9314 (10%)
Combined bacteraemia and fungaemia	13 (0.04%)	12 (0.03%)	25 (0.03%)	18 (0.05%)	32 (0.06%)	50 (0.05%)

All numbers are: Total number; Positive for any fungi (%positive) unless stated otherwise. * Because of a policy change, all blood cultures in the capital region of Denmark were changed to the BACTEC™ system in 2020.

**Table 3 jof-09-00441-t003:** Overview of candidemia’s in the cohort. Absolute difference and numbers needed to treat.

Overall Comparison						
	2018		2020			
	Fungal blood infections	Blood culture sets	Fungal blood infections	Blood culture sets	Absolute difference (95% CI), pr. 1.000 blood culture sets	Numbers needed to treat (95% CI)
Overall estimates	150	83,159	263	92,257	1.2 (0.72; 1.6)	853 (617; 1382)
Stratified analyses						
High-risk departments	89	164	186	18,489	5.2 (3.4; 7.1)	191 (142; 293)
Gastric surgery	13	3617	39	4044	6.8 (3.2; 10)	147 (97; 311)
Gastroenterology	21	224	18	2126	−0.93 (−6.5; 4.6)	-
Hematology	2	1701	10	1787	4.8 (0.98; 8.6)	209 (116; 1024)
Intensive care unit	36	2899	94	4986	7.4 (1.9; 13)	136 (78; 532)
Nephrology	4	2473	10	2181	3.2 (−0.03; 6.4)	311 (155; Inf)
Oncology	5	247	2	2444	−1.2 (−3.3; 0.87)	-
Urology	8	1	13	921	7.4 (−2; 17)	136 (60; Inf)
Low-risk departments	61	66,759	77	73,768	0.16 (−0.17; 0.48)	6367 (2067; Inf)
BACTEC™						
	2018		2020			
	Fungal blood infections	Blood culture sets	Fungal blood infections	Blood culture sets	Absolute difference (95% CI), pr. 1.000 blood culture sets	Numbers needed to treat (95% CI)
Overall estimates	94	34,977	123	37,724	0.62 (−0.17; 1.4)	1609 (709; Inf)
Stratified analyses						
High-risk departments	60	1043	97	1176	2.7 (0.56; 4.9)	364 (203; 1780)
Gastric surgery	9	925	16	1166	4.2 (−5; 13)	238 (75; Inf)
Gastroenterology	16	142	9	1305	−4.9 (−12; 2.2)	-
Hematology	2	1697	10	1783	4.8 (0.98; 8.6)	208 (116; 1020)
Intensive care unit	16	1221	40	2634	2 (−5.9; 9.8)	503 (102; Inf)
Nephrology	4	1967	7	1717	2.1 (−1.5; 5.7)	468 (174; Inf)
Oncology	5	22	2	2235	−1.4 (−3.8; 0.92)	-
Urology	8	1	13	920	−9.3 (−15; −3.8)	-
Low-risk departments	34	24,547	26	25,964	−0.42 (−1; 0.19)	-
Hospitals that used BacTAlert™ in 2018. This changed to BACTEC™ in 2020						
	2018		2020			
	Fungal blood infections	Blood culture sets	Fungal blood infections	Blood culture sets	Absolute difference (95% CI), pr. 1.000 blood culture sets	Numbers needed to treat (95% CI)
Overall estimates	56	48,182	140	54,533	1.6 (1; 2.1)	637 (478; 953)
Stratified analyses						
High-risk departments	29	597	89	6729	9.7 (6.4; 13)	103 (78; 155)
Gastric surgery	4	2692	23	2878	7.3 (3.8; 11)	137 (92; 266)
Gastroenterology	5	820	9	821	5.5 (−3.4; 14)	183 (70; Inf)
Hematology	0	4	0	4	0 (-)	-
Intensive care unit	20	1678	54	2352	13 (5.5; 21)	75 (47; 182)
Nephrology	0	506	3	464	7.1 (−0.12; 14)	140 (70; Inf)
Oncology	0	270	0	209	0 (-)	-
Urology	NA	NA	0	1	-	-
Low-risk departments	27	42,212	51	47,804	0.48 (0.1; 0.86)	2075 (1161; 9717)

## Data Availability

Due to data protection laws, we unfortunately cannot share data.

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
