# Peer review of "A Dedicated Mycosis Flask Increases the Likelihood of Identifying Candidemia Sepsis"

_jof, 2023, doi:10.3390/jof9040441_

Round 1

Reviewer 1 Report

Congratulations to the authors for such a nice work.

Authors may want to include as references international guidelines on how to treat candidemia and discuss their results with regards to such publications. Additionally, authors may also want to confront their data with the EQUAL Score cards for Candida, which may support their results and conclusions.

Author Response

Comment

Congratulations to the authors for such a nice work.

Reply

Thank you very much

Comment

Authors may want to include as references international guidelines on how to treat candidemia and discuss their results with regards to such publications. Additionally, authors may also want to confront their data with the EQUAL Score cards for Candida, which may support their results and conclusions.

Reply

We thank the reviewer for suggesting to add the EQUAL score the presented data. However, we do not have clinical data available, only data that is available via the laboratory information system. The approval for the current study only permits us to extract these data.

We have added and discussed our study in the context of international guidelines specifically with regards to the EQUAL. We have added the following (page xx, line xx)

“Finally, we did not have any clinical data available and therefore could not evaluate whether the intervention was associated with a more favorable outcome e.g. with regards to the EQUAL score (16).”

Reviewer 2 Report

In the article "A dedicated mycosis flask increases the likelihood of identifying candidemia sepsis", the authors analyze whether including the mycosis flask in a set of blood cultures increases the probability of detecting patients with candidemia. They also evaluate whether the inclusion of the mycosis flask differs in high-risk departments compared to low-risk departments, also whether the change in the blood culture system from BacTAlert™ (bioMerieux, France) to BACTEC™ (BD, Ontario, Canada) affects the probability of identifying candidemia. It is a well-written article, and this study also has the strength that it included a large number of samples (more than 700,000 blood culture bottles), which allowed obtaining results with sufficient strength. In addition, I consider that it is a guide for clinical laboratories to decide which methods are used according to their needs and opportunities. I only have one comment:

In stratified analyzes of hospitals using BacTAlert™ vs. BACTEC™ in 2018, we found that the BACTEC™ blood culture system had a substantially higher positivity rate than the BacTAlert™ system, implying that the BACTEC™ system is better than the BacTAlert™ system for identifying candidemia, it would be interesting for you to discuss why the BACTEC™ system detected fungi better than the Bac-TAlert™ system. 

On the other hand, it would be convenient to broaden the discussion to explain their findings since they showed that the inclusion of a flask for mycoses in all sets of blood cultures increased the probability of identifying candidemia in patients from high-risk departments, mainly, however, What would be the reason that this effect was not so evident in patients from departments without high risk?

Author Response

Reviewer 2

Comment
In the article "A dedicated mycosis flask increases the likelihood of identifying candidemia sepsis", the authors analyze whether including the mycosis flask in a set of blood cultures increases the probability of detecting patients with candidemia. They also evaluate whether the inclusion of the mycosis flask differs in high-risk departments compared to low-risk departments, also whether the change in the blood culture system from BacTAlert™ (bioMerieux, France) to BACTEC™ (BD, Ontario, Canada) affects the probability of identifying candidemia. It is a well-written article, and this study also has the strength that it included a large number of samples (more than 700,000 blood culture bottles), which allowed obtaining results with sufficient strength. In addition, I consider that it is a guide for clinical laboratories to decide which methods are used according to their needs and opportunities. I only have one comment:

In stratified analyzes of hospitals using BacTAlert™ vs. BACTEC™ in 2018, we found that the BACTEC™ blood culture system had a substantially higher positivity rate than the BacTAlert™ system, implying that the BACTEC™ system is better than the BacTAlert™ system for identifying candidemia, it would be interesting for you to discuss why the BACTEC™ system detected fungi better than the Bac-TAlert™ system. 

Reply

This is a very good question. We have some studies that support the results in the current study. We have added this to the discussion (We have added the following page xx, line xx)

“, one comparative study found that BACTEC™ system had statistically significantly lower TTD than BacTAlert™ system for the detection of fungi which supports the finding in the current study (10).

Comment

On the other hand, it would be convenient to broaden the discussion to explain their findings since they showed that the inclusion of a flask for mycoses in all sets of blood cultures increased the probability of identifying candidemia in patients from high-risk departments, mainly, however, What would be the reason that this effect was not so evident in patients from departments without high risk?

Reply

This is in our opinion a very interesting. One would expect that the effect of such an intervention would be distributed equally among all patients. However, the findings in this study suggests statistical interaction i.e. a different effect depending on the risk of the patient, we were not able to think of a plausible explanation for this we have discussed this in some more detail (page xx, line xx)

“, implying statistical interaction. Whether or not this is a chance finding or indeed a phenomenon that can reproduced, needs to be evaluated in future studies”